# Structure Guiding Supramolecular Assemblies in Metal-Organic Multi-Component Compounds of Mn(II): Experimental and Theoretical Studies

**Manjit K. Bhattacharyya** [1,*], **Kamal K. Dutta** [1], **Pranay Sharma** [1], **Rosa M. Gomila** [2], **Miquel Barceló-Oliver** [2] and **Antonio Frontera** [2,*]

1   Department of Chemistry, Cotton University, Guwahati 781001, Assam, India
2   Departament de Química, Universitat de les Illes Balears, Crta de Valldemossa km 7.7, 07122 Palma de Mallorca, Balears, Spain
*   Correspondence: manjit.bhattacharyya@cottonuniversity.ac.in (M.K.B.); toni.frontera@uib.es (A.F.)

**Abstract:** Two multi-component coordination compounds of Mn(II), viz. $[Mn(H_2O)_6](2\text{-Mepy})_2(4\text{-}NO_2bz)_2 \cdot 2H_2O$ (**1**) and $[Mn(H_2O)_6][Mn(2,3\text{-PDCH})_3]_2$ (**2**) (where, 2-Mepy = 2-methylpyridine, 4-$NO_2bz$ = 4-nitrobenzoate, 2,3-PDC = 2,3-pyridinedicarboxylate), have been synthesized and characterized using elemental, spectroscopic (FT-IR and electronic), TGA and single-crystal X-ray diffraction analyses. Complex **1** is a co-crystal hydrate of Mn(II) involving uncoordinated 2-Mepy, 4-$NO_2bz$ and water molecules; while compound **2** is a multi-component molecular complex salt of Mn(II) comprising cationic $[Mn(H_2O)_6]^{2+}$ and anionic $[Mn(2,3\text{-PDCH})_3]^-$ complex moieties. The uncoordinated 2-Mepy and 4-$NO_2bz$ moieties of **1** are involved in lone-pair (l.p)-$\pi$ and C–H$\cdots\pi$ interactions which stabilize the layered assembly of the compound. The crystal structure of compound **2** has been previously reported. However, we have explored the unusual enclathration of complex cationic moieties within the supramolecular host cavities formed by the molecular assembly of complex anionic moieties. The supramolecular assemblies obtained in the crystal structure have been further studied theoretically using DFT calculations, quantum theory of atoms-in-molecules (QTAIM) and non-covalent interaction plot (NCI plot) computational tools. Theoretical studies reveal that the combination of $\pi$-staking interactions (l.p-$\pi$, $\pi$-$\pi$ and C–H$\cdots\pi$) have more structure-guiding roles compared to the H-bonds. The large binding energy of $\pi$-stacking interactions in **2** is due to the antiparallel orientation of aromatic rings and their coordination to the metal centers, thereby increasing the contribution of the dipole–dipole interactions.

**Keywords:** metal-organic; multi-component; co-crystals; supramolecular; enclathration; DFT calculations





## 1. Introduction

Metal-organic compounds of transition metals have attracted the researchers not only because of their intriguing structural topologies but also due to their myriad applications in various fields [1–4]. Using organic moieties as ligands, various research groups have reported coordination compounds with fascinating topologies and network architectures [5]. Multi-component compounds (composed of more than two components), mainly classified as co-crystals, molecular salts and polymorphs, have also received remarkable emphasis because of their significant physicochemical and pharmaceutical applications [6,7]. The co-crystallization technique has been effectively employed in the literature to develop potential co-crystals with significant pharmaceutical properties [8]. Metal–organic multi-component co-crystals have also held a unique place in crystal engineering because of their applications in pharmaceuticals, electronic devices and in synthetic chemistry [9,10].

Supramolecular chemistry deals with non-covalent interactions, which mainly include HB, stacking, charge transfer interactions, etc. [11,12]. During molecular self-assembly, weak non-covalent interactions, viz. anion-$\pi$, cation-$\pi$, $\pi$-stacking, C–H/$\pi$, $\sigma/\pi$-hole,

lone-pair/$\pi$, etc., play decisive roles in the stability of the compounds owing to their combined strength, directionality and ability to act synergistically [13–15]. The cooperative interplay of $\pi$-stacking interactions also attracts researchers from the crystal engineering viewpoint [16,17]. The inclusion of self-assembled guests within the host cavities in supramolecular architectures usually depends on the molecular association and the size of the molecules [18].

The uses of ancillary N-donors with aromatic carboxylate ligands have drawn considerable emphasis in the design and synthesis of metal–organic coordination compounds [19,20]. Researchers often employed aromatic carboxylates as ligands for the synthesis of compounds due to their multi-coordination modes for transition metals [21,22]. Pyridine dicarboxylate (PDC) derivatives have also been successfully employed as building blocks to construct metal–organic compounds of fascinating structural topologies [23,24]. A 2-methylpyridine (2-Mepy) moiety has also been used to construct metal–organic compounds with interesting structural topologies [25,26].

In the present study, we have reported the synthesis and crystal structures of two Mn(II) multi-component coordination compounds, namely, $[Mn(H_2O)_6](2\text{-Mepy})_2(4\text{-NO}_2bz)_2 \cdot 2H_2O$ (**1**) and $[Mn(H_2O)_6][Mn(2,3\text{-PDCH})_3]_2$ (**2**). The compounds have been characterized using various spectroscopic and analytical techniques. Compound **1** is a Mn(II) co-crystal hydrate having uncoordinated 2-Mepy, 4-NO$_2$bz and water molecules, while compound **2** is a multi-component complex molecular salt of Mn(II) involving cationic $[Mn(H_2O)_6]^{2+}$ and anionic $[Mn(2,3\text{-PDCH})_3]^-$ moieties. Structural investigations of **1** unfold the presence of C–H$\cdots\pi$ and l.p-$\pi$ contacts with the uncoordinated 2-Mepy and 4-NO$_2$bz moieties. An unusual enclathration of the guest complex cationic $[Mn(H_2O)_6]^{2+}$ moiety within the hexameric supramolecular host cavities formed by the anionic $[Mn(2,3\text{-PDCH})_3]^-$ moieties in compound **2** stabilizes the supramolecular assemblies. We have performed computational studies to analyze the supramolecular assemblies of the crystal structures, paying special attention to the $\pi$-stacking interactions in the compounds. The characteristics and the energetic features of the supramolecular assemblies of the compounds have been explored using DFT calculations, NCI plotting and QTAIM computational tools.

## 2. Experimental Section

### 2.1. Materials and Methods

Manganese(II) chloride tetrahydrate, 2-methylpyridine, 4-nitrobenzoic acid, 2,3-pyridine dicarboxylic acid were bought from Sigma Aldrich (Darmstadt, Germany) and Merck (Darmstadt, Germany) and used without further purification. Elemental analyses of the compounds were recorded in a Perkin Elmer 2400 Series II CHNS/O analyzer (Waltham, MA, USA). A Bruker ALPHA II Infrared spectrophotometer (West Perth, WA, Australia) was used to record the FTIR spectra of the compounds. The electronic spectra of the compounds were recorded in a Shimadzu UV-2600 spectrophotometer (Duisburg, Germany). To record the UV-Vis-NIR spectra, BaSO$_4$ was used as reference. A Sherwood Mark 1 Magnetic Susceptibility balance(Cambridge, UK) was used to calculate the magnetic susceptibility of the compounds. Thermogravimetric analyses of the compounds were performed in a Mettler Toledo TGA/DSC1 STAR$^e$ system (Columbus, OH, USA) under the flow of N$_2$ gas at a heating rate of 10 °C min$^{-1}$.

### 2.2. Synthesis

#### 2.2.1. Synthesis of $[Mn(H_2O)_6](2\text{-Mepy})_2(4\text{-NO}_2bz)_2 \cdot 2H_2O$ (**1**)

MnCl$_2$·4H$_2$O (0.197 g, 1 mmol) and 2-Mepy (0.2 mL, 2 mmol) were mixed in de-ionised water (10 mL) and mechanically stirred at room temperature for two hours. After two hours, sodium salt of 4-nitrobenzoate (0.334 g, 2 mmol) was added and stirring was continued for another hour (Scheme 1). The reaction mixer was then kept in a refrigerator for crystallization. After several days, block colorless crystals suitable for SCXRD were obtained. Yield: 0.602 g (84%). Anal. calcd. for C$_{26}$H$_{38}$MnN$_4$O$_{16}$: C, 43.52%; H, 5.34%; N, 7.81%; Found: C, 43.41%; H, 5.29%; N, 7.72%. IR spectral data (KBr disc, cm$^{-1}$): 3449(br),

2923(w), 2839(w), 2730(w), 1962(w), 1595(m), 1579(s), 1520(m), 1386(m), 1344(m), 1161(w), 1111(w), 1044(w), 1002(w), 878(sh), 811(s), 752(m), 719(m), 627(w) (s, strong; m, medium; w, weak; br, broad; sh, shoulder).

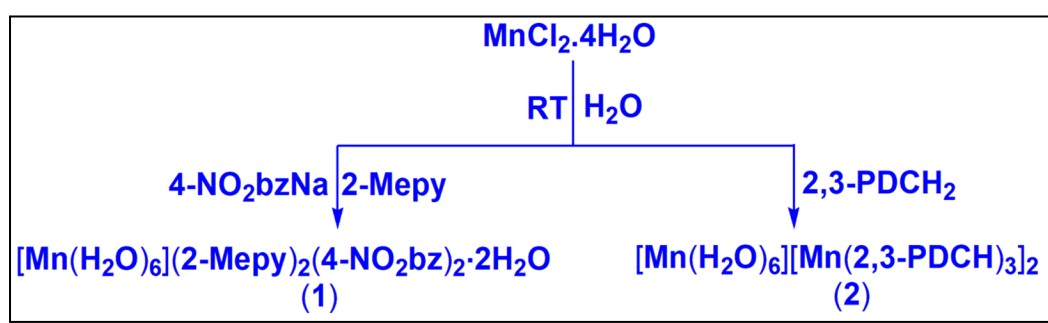

**Scheme 1.** Synthesis of the compounds **1** and **2**.

### 2.2.2. Synthesis of $[Mn(H_2O)_6][Mn(2,3\text{-PDCH})_3]_2$ (**2**)

$MnCl_2 \cdot 4H_2O$ (0.197 g, 1 mmol) and 2,3-pyridine dicarboxylic acid (0.334 g, 2 mmol) were dissolved in 10 mL of distilled water and mechanically stirred at room temperature for two hours (Scheme 1). The pale yellow solution was kept undisturbed in cooling conditions for crystallization, which yields block-shaped yellow-colored crystals after a few days. Yield: 1.053 g (83%). Anal. calcd. for: $C_{42}H_{36}Mn_3N_6O_{30}$: C, 39.73%; H, 2.86%; N, 6.62%; Found: C, 39.66%; H, 2.81%; N, 6.51%. IR spectral data (KBr disc, $cm^{-1}$): 3432(m), 2539(w), 1704(w), 1629(sh), 1579(sh), 1371(m), 1386(m), 1252(m), 1136(m), 1111(s), 885(m), 836(m), 702(w), 677(w).

### 2.3. Crystallographic Data Collection and Refinement

X-ray crystallographic data of the compounds were taken on a Bruker D8 Venture diffractometer (New York, NY, USA) containing a Photon III 14 detector, using an Incoatec high-brilliance IµS DIAMOND Cu tube. Data reduction, cell refinements as well as scaling and absorption corrections were performed using the Bruker APEX4 and SADABS program [27]. Crystal structures were solved by the direct method and refined by full-matrix least-squares techniques with SHELXL-2018/3 [28] using WinGX [29]. All non-hydrogen atoms were refined with anisotropic thermal parameters by full-matrix least-squares calculations on $F^2$. Hydrogen atoms were inserted at calculated positions and refined as riders, except for those from water molecules, which were located using a Fourier difference map. Diamond 3.2 software was used to draw the structural diagrams [30]. Table 1 contains the crystallographic data for the compounds.

**Table 1.** Crystallographic data and structure refinement details for the compounds **1** and **2**.

| Crystal Parameters | 1 | 2 |
|---|---|---|
| Empirical formula | $C_{26}H_{38}MnN_4O_{16}$ | $C_{42}H_{36}Mn_3N_6O_{30}$ |
| Formula weight | 717.54 | 1269.59 |
| Temperature (K) | 100.0 | 296.0 |
| Wavelength (Å) | 1.54178 | 0.71073 |
| Crystal system | Triclinic | Trigonal |
| Space group | $P\overline{1}$ | $P\overline{3}$ |
| $a/\text{Å}$ | 7.2732(2) | 14.6618(8) |
| $b/\text{Å}$ | 7.3914(2) | 14.6618(8) |
| $c/\text{Å}$ | 16.6144(4) | 6.3551(4) |
| $\alpha$ ° | 96.3990(1) | 90 |
| $\beta$ ° | 92.4650(1) | 90 |
| $\gamma$ ° | 111.6260(1) | 120 |
| Volume ($\text{Å}^3$) | 821.71(4) | 1183.12(1) |
| Z | 1 | 1 |
| Calculated density ($g/cm^3$) | 1.450 | 1.782 |
| Absorption coefficient ($mm^{-1}$) | 3.957 | 0.901 |

**Table 1.** *Cont.*

| Crystal Parameters | 1 | 2 |
|---|---|---|
| F(000) | 375.0 | 645.0 |
| Crystal size (mm$^3$) | $0.35 \times 0.21 \times 0.18$ | $0.31 \times 0.22 \times 0.12$ |
| $\theta$ range for data collection(°) | 10.76 to 136.33 | 1.604 to 23.973 |
| Index ranges | $-8 <= h <= 8$, $-8 <= k <= 8$, $-19 <= l <= 19$ | $-16 <= h <= 16$, $-16 <= k <= 16$, $-7 <= l <= 7$ |
| Reflections collected | 19,804 | 28,974 |
| Unique data ($R_{int}$) | 2972 | 1249 |
| Refinement method | Full-matrix least-squares on F$^2$ | Full-matrix least-squares on F$^2$ |
| Data/restraints/parameters | 2972/2/225 | 1249/0/123 |
| Goodness-of-fit on $F^2$ | 1.083 | 1.055 |
| Final $R$ indices[$I > 2\sigma$ ($I$)] $R1/wR2$ | 0.0249/0.0691 | 0.0227/0.0574 |
| $R$ indices(all data) $R1/wR2$ | 0.0249/0.0691 | 0.0254/0.0590 |
| Largest diff. peak and hole (e.Å$^{-3}$) | 0.27 and $-0.46$ | 0.203 and $-0.266$ |

CCDC 2,123,024 and 2,123,025 contain the supplementary crystallographic data for the compounds **1** and **2**, respectively. These data can be obtained free of charge at http://www.ccdc.cam.ac.uk (accessed on 13 July 2021) or from the Cambridge Crystallographic Data Centre, 12 Union Road, Cambridge CB2 1EZ, UK; fax: (+44) 1223-336-033; or E-mail: deposit@ccdc.cam.ac.uk.

*2.4. Computational Methods*

The RI-BP86-D3/def2-TZVP [31,32] level of theory was used to compute the assemblies of compounds **1** and **2** using the X-ray coordinates and the Turbomole 7.2 program [33]. The noncovalent interaction plot (NCIPlot) [34] and the quantum theory of "atoms-in-molecules" (QTAIM) [35] computational methods were computed using the MULTIWFN program [36] and represented by means of the VMD software [37].

## 3. Results and Discussion

*3.1. Syntheses and General Aspects*

[Mn(H$_2$O)$_6$](2-Mepy)$_2$(4-NO$_2$bz)$_2$·2H$_2$O (**1**) is obtained by reacting one equivalent of MnCl$_2$·4H$_2$O, two equivalents of 2-Mepy with two equivalents of Na-4-NO$_2$bz at ambient conditions in water medium. Similarly, [Mn(H$_2$O)$_6$][Mn(2,3-PDCH)$_3$]$_2$ (**2**) has been isolated by reacting MnCl$_2$·4H$_2$O and 2,3-pyridinedicarboxylic acid at a 1:2 molar ratio at room temperature in water medium. The compounds are soluble in water and in common organic solvents. Compounds **1** and **2** exhibit room temperature (298 K) magnetic moments ($\mu_{eff}$) of 5.87 and 5.81 BM, respectively, which reveals the presence of five unpaired electrons per Mn(II) center [38]. Although the crystal structure of the compound 2 has been previously reported [1], we have investigated the role of non-covalent interactions and the unusual enclathration of the complex cationic moieties during molecular association.

*3.2. Crystal Structure Analysis*

The molecular structure of compound **1** has been depicted in Figure 1. Table 2 represents the bond lengths and bond angles around the Mn(II) center. Compound **1** crystallizes in the triclinic P$\bar{1}$ space group. The Mn(II) center is found to be present in a crystallographic center of inversion. The Mn(II) center has an octahedral geometry having six coordinated water molecules (O1W, O1W′, O2W, O2W′, O3W and O3W′). Moreover, two uncoordinated 2-Mepy, two uncoordinated 4-NO$_2$bz and two uncoordinated water molecules are also present. The equatorial sites around the Mn(II) center are occupied by four oxygen atoms (O1W, O1W′, O2W, O2W′) from four coordinated water molecules, whereas the axial positions are filled by O3W and O3W′ water molecules. The Mn–O$_{axial}$ bond lengths (2.178 Å) are slightly higher than that of Mn–O$_{equitorial}$ bond lengths (2.156 Å). The average Mn–O bond lengths are well consistent with those reported for Mn(II) compounds [39].

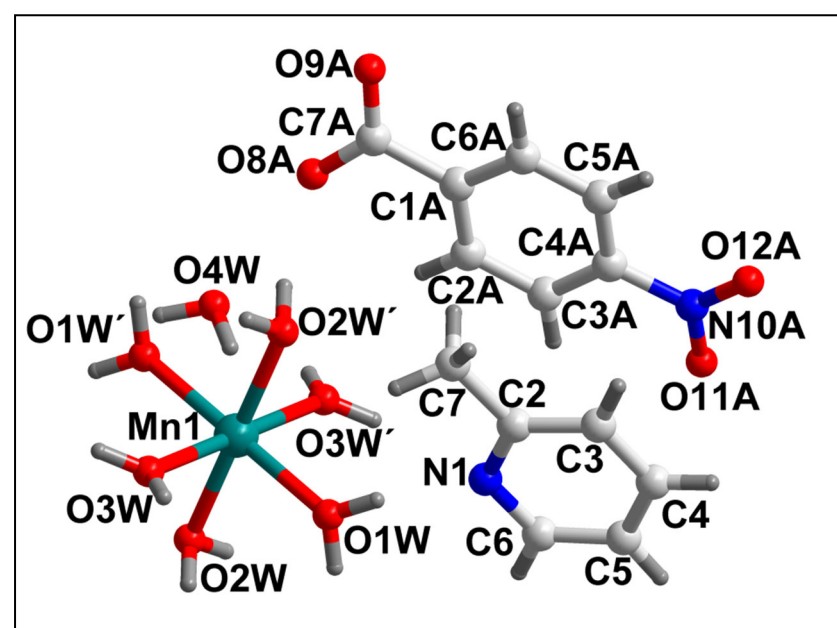

**Figure 1.** Molecular structure of [Mn(H$_2$O)$_6$] 2(2-Mepy) 2(4-NO$_2$bz)·2H$_2$O (**1**).

**Table 2.** Selected bond lengths (Å) and bond angles (°) around the Mn(II) centers in **1** and **2**, respectively.

| Bond Lengths of 1 (in Å) | | Bond Angles of 1 (in Deg) | |
|---|---|---|---|
| Mn1–O1W#1 | 2.1514 | O1W#1–Mn1–O1W | 180.0 |
| Mn1–O1W | 2.1515 | O1W–Mn1–O3W | 88.1 |
| Mn1–O3W#1 | 2.1782 | O1W–Mn1–O3W#1 | 91.9 |
| Mn1–O3W | 2.1782 | O1W#1–Mn1–O3W | 91.9 |
| Mn1–O2W | 2.1605(9) | O1W#1–Mn1–O3W#1 | 88.1 |
| Mn1–O2W#1 | 2.1605(9) | O1W#1–Mn1–O2W#1 | 89.21(3) |
| | | O1W–Mn1–O2W | 89.20(3) |
| | | O1W#1–Mn1–O2W | 90.80(3) |
| | | O1W–Mn1–O2W#1 | 90.79(3) |
| | | O3W#1–Mn1–O3W | 180.0 |
| | | O2W–Mn1–O3W#1 | 88.14(4) |
| | | O2W#1–Mn1–O3W#1 | 91.86(4) |
| | | O2W–Mn1–O3W | 91.86(4) |
| | | O2W#1–Mn1–O3W | 88.14(4) |
| | | O2W#1–Mn1–O2W | 180.0 |
| Bond lengths of **2** (in Å) | | Bond angles of **2** (in Degree) | |
| Mn1–O3 | 2.1382(1) | O3–Mn1–O3 | 95.0(5) |
| Mn1–N1 | 2.2758(1) | O3–Mn1–N1 | 100.59(5) |
| Mn2–O5 | 2.1678(1) | O3–Mn1–N1 | 74.23(5) |
| | | O3–Mn1–N1 | 161.68(5) |
| | | N1–Mn1–N1 | 92.91(5) |
| | | O5–Mn2–O5 | 180.00(8) |
| | | O5–Mn2–O5 | 93.30(6) |
| | | O5–Mn2–O5 | 86.70(6) |

#1 1-X, 1-Y, 1-Z.

The cationic complex moieties of compound **1** viz. [Mn(H$_2$O)$_6$]$^{2+}$ are interconnected via the lattice O4W water molecule along the crystallographic a direction (Figure S1). The coordinated and uncoordinated water molecules are involved in O–H···O interactions with O1W–H1WB···O4W and O2W–H2WA···O4W distances of 1.97 and 1.92 Å, respectively (Table 3).

**Table 3.** HB parameters (Å and °) of compounds **1** and **2**.

| D–H···A | d(D–H) | d(D–A) | d(H···A) | <(DHA) |
|---|---|---|---|---|
| **Compound 1** | | | | |
| O2W–H2WA···O4W | 0.87 | 2.732(1) | 1.92 | 153.8 |
| O1W–H1WB···O4W | 0.86 | 2.835(2) | 1.97 | 170.1 |
| C3–H3···O12A | 0.95 | 3.490(2) | 2.63 | 150.7 |
| C4–H4···O11A | 0.95 | 3.449(2) | 2.61 | 147.2 |
| C5–H5···O12A | 0.95 | 3.422(2) | 2.59 | 146.2 |
| C7–H7B···C6 | 0.97 | 3.708(2) | 3.26 | 109.1 |
| C7–H7A···C6 | 0.98 | 3.708(2) | 3.30 | 106.7 |
| O4W–H4WB···O8A | 0.86 | 2.805(1) | 1.93 | 177.6 |
| O4W–H4WB···O9A | 0.87 | 2.797(2) | 1.93 | 170.3 |
| O1W–H1WA···N1 | 0.87 | 2.749(1) | 1.88 | 173.3 |
| O4W–H4WA···O8A | 0.87 | 2.771(1) | 1.91 | 169.5 |
| **Compound 2** | | | | |
| O5–H5A···O4 | 0.82 | 3.050(2) | 2.28 | 154.6 |
| O5–H5B···O1 | 0.92 | 2.851(2) | 1.95 | 162.3 |
| C3–H3···O3 | 0.93 | 3.755(2) | 2.94 | 146.5 |
| O2–H2···O4 | 0.95 | 2.584(2) | 1.64 | 166.1 |

Figure 2 represents a 2D assembly of compound **1** along the crystallographic ac plane assisted by O–H···O, C–H···O, O–H···N and non-covalent C–H···C interactions. The water molecules (O3W and O4W) are involved in strong O–H···O hydrogen bonding (HB) interactions with the carboxylate O-atom (O8A and O9A) of 4-NO$_2$bz having O3W–H3WB···O8A and O4W–H4WB···O9A bond distances of 1.93 and 1.93 Å, respectively. C–H···O HB interaction is also observed, involving the –CH moieties of 2-Mepy and O-atoms of 4-NO$_2$bz having C3–H3···O12A, C4–H4···O11A and C5–H5···O12A distances of 2.63, 2.61 and 2.59 Å, respectively. The coordinated water molecule (O1W) is involved in O–H···N HB interactions with N1 atom of 2-Mepy having a O1W–H1WA···N1 distance of 1.883 Å. Moreover, weak bifurcated C–H···C interactions are also present in the crystal structure involving the methyl groups of 2-Mepy [C7–H7B···C6 = 3.26 Å, C(sp$^3$)–H···C(sp$^2$), C7···C6 = 3.708 Å, C7–H7A···C6 = 3.30 Å, C(sp$^3$)–H···C(sp$^2$), C7···C6 = 3.708 Å] [40].

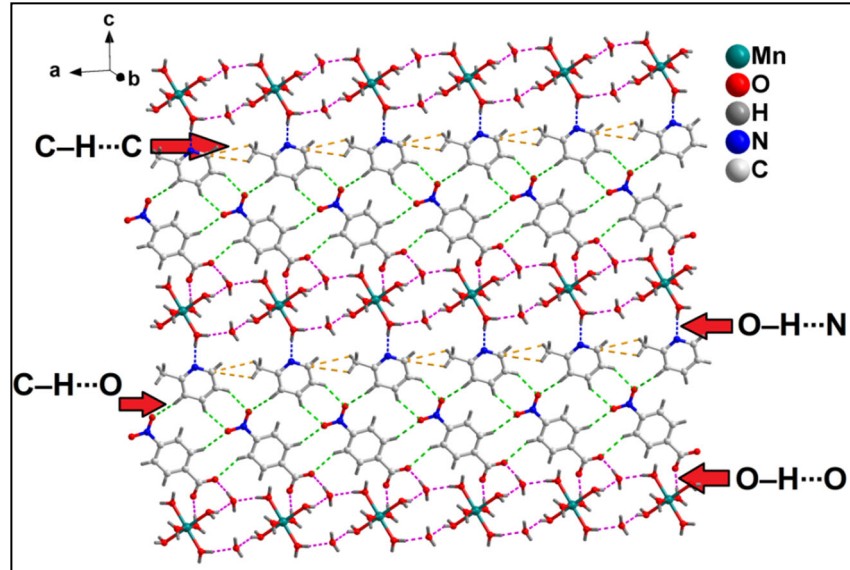

**Figure 2.** 2D network architectures of **1** along the crystallographic ac plane assisted by O–H···O, C–H···O, O–H···N HB and non-covalent C–H···C interactions.

The lattice 2-Mepy, 4-NO$_2$bz and water molecules of compound **1** are aggregated to form the layered assembly via non-covalent O–H···O, C–H···O, π-π, C–H···π and lone-pair

(l.p.)···π interactions (Figure 3). O–H···O interactions between the O4W and carboxylate O-atoms (O8A and O9A) of lattice 4-NO$_2$bz are observed having O4W–H4WB···O8A and O4W–H4WA···O9A bond distances of 1.91 and, 1.93 Å, respectively. The –CH groups of 2-Mepy are involved in C–H···O HB interactions with oxygen atom (O11A and O12A) of 4-NO$_2$bz having C4–H4···O11A and C5–H5···O12A distances of 2.61 and 2.59 Å, respectively. Another C–H···O HB interaction is also observed between the –CH moiety of the methyl group of 2-Mepy and O atom (O9A) of the carboxylate group of lattice 4-NO$_2$bz having a C7–H7C···O9A distance of 2.96 Å. An aromatic π-stacking interaction is observed in the layered assembly involving the aromatic rings of 2-Mepy and 4-NO$_2$bz having a centroid(N1, C2–C6)–centroid(C1A–C6A) separation of 3.862 Å. The slipped angle (the angle formed by the ring normal and the vector between the two ring centroids) of the two aromatic rings is found to be 20.6°, which is close to the literature value [41]. Moreover, a non-covalent C–H···π interaction is also observed involving the –C7H7A moiety of lattice 2-Mepy and π-electrons of the aromatic ring of lattice 4-NO$_2$bz. The C7···Cg and H7A···Cg distances are found to be 3.697 and 2.845 Å, respectively (Cg is the ring centroid defined by the atoms C1A, C2A, C3A, C4A C5A and C6A). The corresponding angle for the C–H···π interaction is 145.7°, which indicates the significance of the interaction [42]. The O11A atom of the lattice 4-NO$_2$bz is involved in lone-pair (l.p.)···π interactions with the π-electrons of the aromatic ring of lattice 2-Mepy with O11A···Cg distances of 3.785 Å (where Cg is the ring centroid defined by atoms N1, C2, C3, C4, C5, C6). These non-covalent interactions between the uncoordinated 2-Mepy and 4-NO$_2$bz moieties have been further studied theoretically (vide infra).

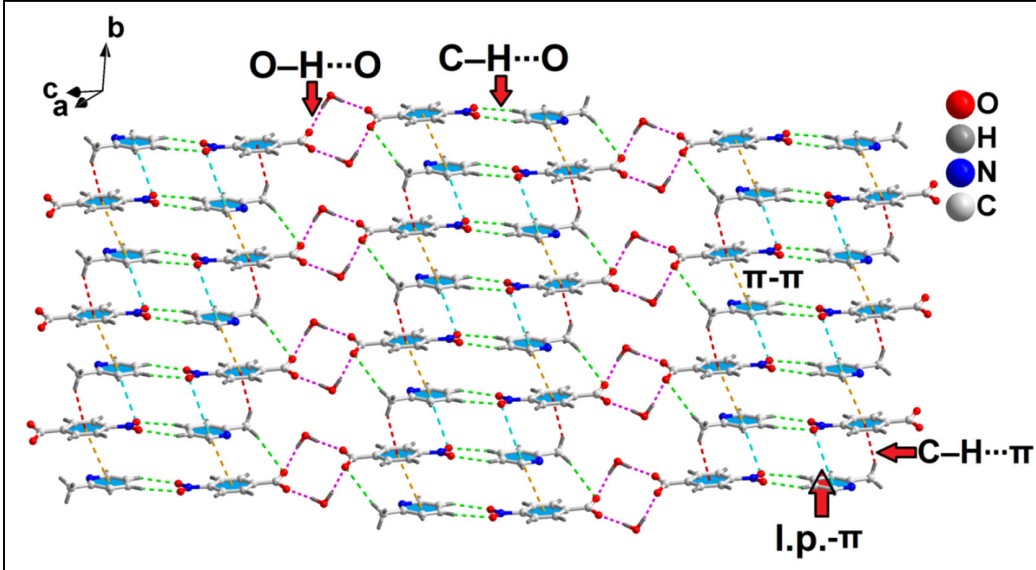

**Figure 3.** Layered assembly of compound **1** involving the lattice 2-Mepy, 4-NO$_2$bz and water molecules.

The molecular structure of compound **2** has been depicted in Figure 4. Table 2 contains the bond lengths and bond angles around the metal centers. Compound **2** crystallizes in the trigonal P3̄ space group. The asymmetric unit of compound **2** comprises a dicationic complex moiety and two anionic complex moieties. Crystal structure analysis reveals that the Mn1 center of compound **2** lies on three-fold symmetry axis, whereas the Mn2 center lies not only on a three-fold axis but also on an inversion center. The average Mn–O and Mn–N bond lengths are consistent with those reported for similar complexes [43,44].

1D supramolecular chain of **2** is stabilized by π-π, O–H···O and C–H···O HB interactions (Figure 5). In the 1D supramolecular chain, two different types of dimers are observed. The homo-dimer is formed between the two complex anionic moieties of **2** involving π-stacking and C–H···O HB interactions. π-stacking interaction 2,3-PDCH moieties are observed, having a centroid(N1, C2-C6)–centroid(N1, C2-C6) distance of 3.832 Å

with a slipped angle of 20.1°. Moreover, C–H···O HB interaction is also observed in the homo-dimer involving the coordinated O atom (O3) of 2,3-PDCH and –CH moiety of a neighboring 2,3-PDCH having C3–H3···O3 distances of 2.94 Å. Similarly, a hetero-dimer is also observed in the 1D chain involving the complex cationic and anionic moieties assisted by O–H···O HB interactions (see Table 3). These homo- and hetero-dimers have been further studied theoretically (vide infra).

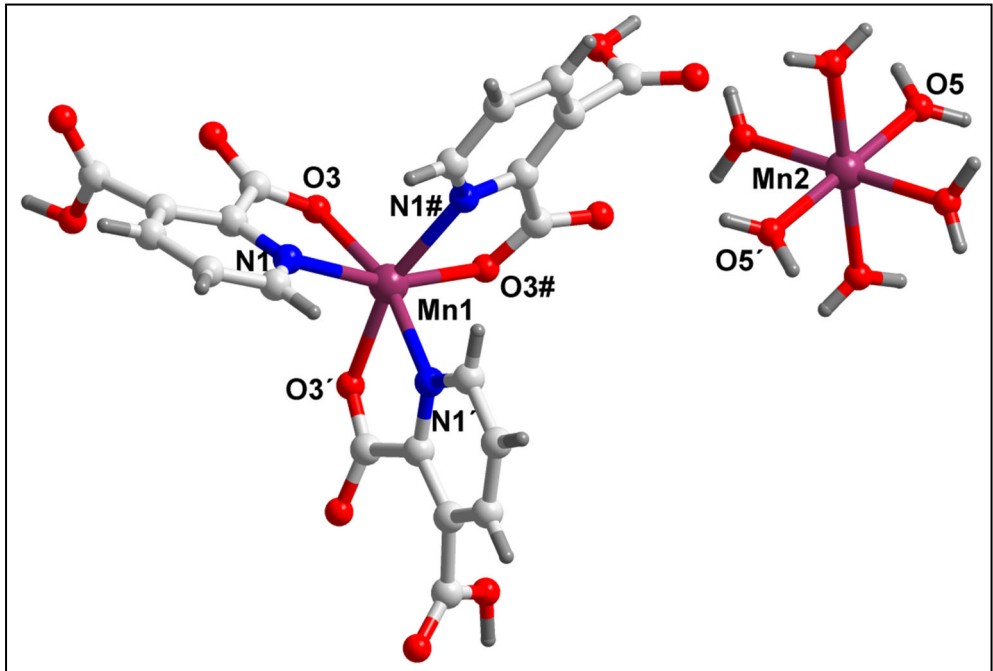

**Figure 4.** Molecular structure of [Mn(H$_2$O)$_6$][Mn(2,3-PDCH)$_3$]$_2$ (**2**).

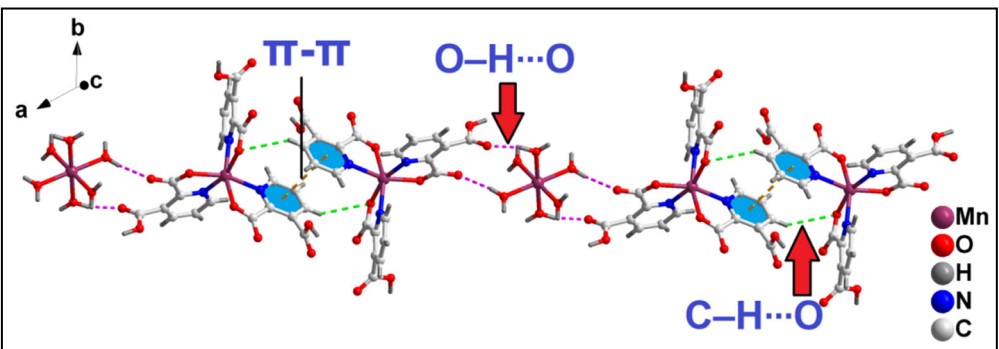

**Figure 5.** 1D supramolecular chain of compound **2** involving aromatic π-stacking, O–H···O and C–H···O HB interactions.

Further analysis unfolds the enclathration of the cationic complex moiety viz. [Mn(H$_2$O)$_6$]$^{2+}$ within a supramolecular hexameric cavity built by the [Mn(2,3-PDCH)$_3$]$^−$ moieties (Figure 6a). The supramolecular hexameric cavity is formed by O–H···O and aromatic π-stacking interactions. An uncoordinated O4 atom of 2,3-PDCH is involved in the O–H···O interaction with an O2–H2···O4 distance of 1.64 Å. π-stacking interaction between the aromatic rings of 2,3-PDCH is observed having a centroid(N1, C2–C6)–centroid(N1, C2–C6) distance of 3.832 Å. These enclathrated guest cationic complex moieties stabilize the layered assembly of the compound along the crystallographic ab plane (Figure 6b).

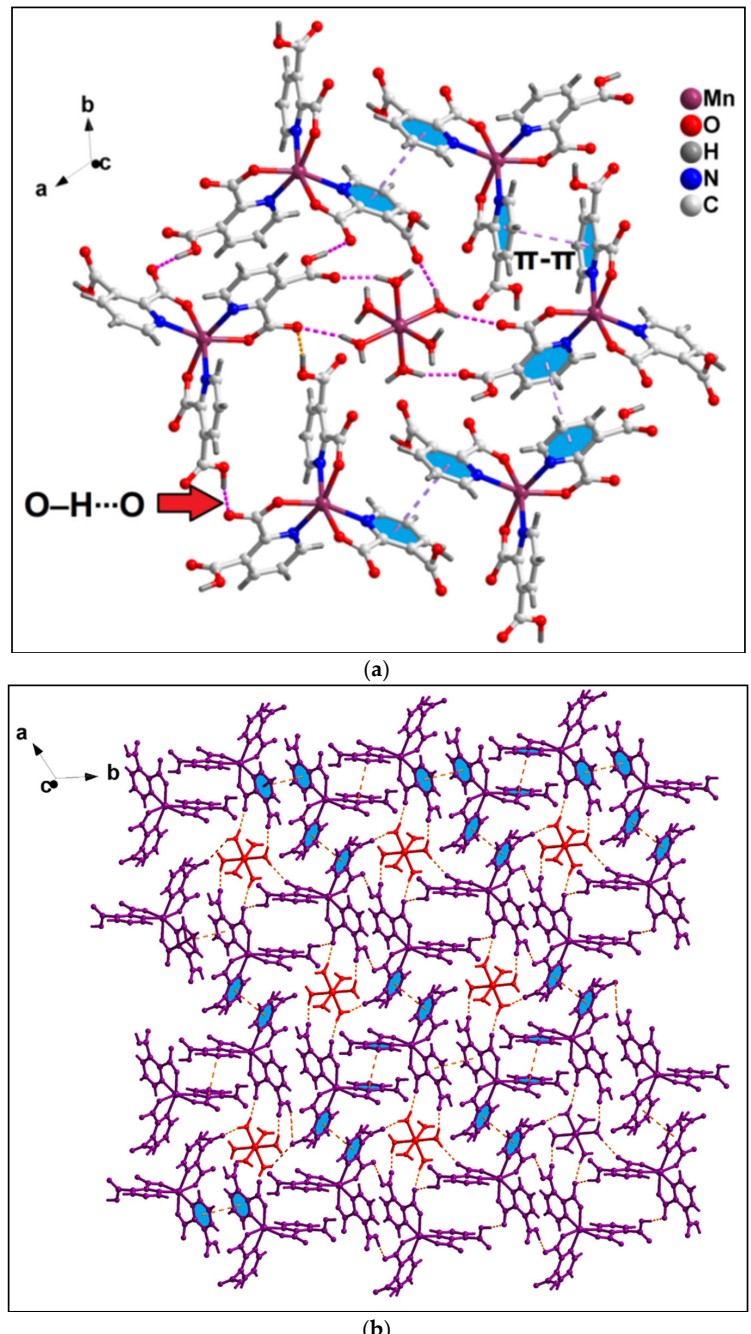

**Figure 6.** (**a**) Enclathrated guest cationic complex moiety of **2** in the supramolecular hexameric cavity formed by the anionic complex moieties; (**b**) 2D architecture of **2** along the crystallographic ab plane with enclathration of guest cationic complex moieties.

### 3.3. Spectral Studies

#### 3.3.1. FT-IR Spectroscopy

FT-IR spectra of the compounds **1** and **2** have been recorded at 4000–500 cm$^{-1}$ (Figure S2). Broad absorption bands at 3449–3415 cm$^{-1}$ are due to the O–H stretching vibrations of the coordinated and/or lattice aqua molecules of **1** and **2** [45]. The presence of bands due to $\rho_r$ (H$_2$O) (716 cm$^{-1}$ in **1** and 710 cm$^{-1}$ in **2**) and $\rho_w$ (H$_2$O) (672 cm$^{-1}$ in **1** and 670 cm$^{-1}$ in **2**) supports the presence of coordinated water molecules in the compounds [46]. The absorption bands at 1579 and 1344 cm$^{-1}$ in **1** are due to the symmetric and asymmetric stretching vibrations of the nitro group of 4-NO$_2$bz, respectively [47,48]. In **1**, $\nu$(C–C) and $\nu$(C–N) stretching frequencies of 2-Mepy are obtained at 1595, 1520 and

1386 cm$^{-1}$, respectively, whereas out-of-plane deformation $\nu$(C–H) stretching vibrations are observed at 811 and 752 cm$^{-1}$, respectively [49]. Lack of bands at around 1710 cm$^{-1}$ in the FT-IR spectrum of **1** suggest the deprotonation of carboxyl groups of 4-NO$_2$bz [50]. The absorption band at 1704 cm$^{-1}$ in **2** indicates that carboxyl moieties of 2,3-PDC ligand are not completely deprotonated on coordination to the Mn(II) center [51]. For compound **2**, the asymmetric $\nu_{as}$(COO$^-$) and symmetric stretching vibrations $\nu_s$(COO$^-$) of carboxylate moieties of 2,3-PDC are obtained at 1629 and 1386 cm$^{-1}$, respectively. The difference [$\Delta\nu = \nu_{as}$(COO$^-$) $- \nu_s$(COO$^-$)] is found to be greater than 200 cm$^{-1}$, corroborating the monodentate coordination of the carboxylate moieties of the 2,3-PDCH to Mn(II) center [52,53].

### 3.3.2. Electronic Spectroscopy

Figures S3 and S4 represent the electronic spectra of the compounds **1** and **2**, respectively. Absence of bands in the visible region is because of the Mn(II) center (d$^5$ system), for which all the electronic transitions from the $^6$A$_{1g}$ ground state are doubly forbidden [54,55]. The absorption bands obtained at 230 and 269 nm in the solid state spectrum of **1** are due to the $\pi\rightarrow\pi^*$ transition of the aromatic ligands [56,57]. However, in the aqueous phase spectrum of **1**, these absorption peaks are obtained at 228 and 271 nm, respectively. The bands at 223 and 268 nm in the UV-Vis-NIR spectrum of **2** can be ascribed due to $\pi\rightarrow\pi^*$ transition of an aromatic ligand [56,57], while these peaks are obtained at 222 and 271 nm, respectively, in the aqueous phase UV-Vis spectrum of **2**.

### 3.4. Thermogravimetric Analysis

The thermogravimetric curves of the compounds have been recorded in the temperature range 25–800 °C under N$_2$ atmosphere at the heating rate of 10 °C/min (Figure S5). For compound **1**, the 17.95% observed weight loss in the temperature range 50–160 °C can be attributed to the decomposition of six coordinated water and two water molecules of crystallization (calcd. = 19.06%) [58]. In the temperature range 161–465 °C, two uncoordinated 4-NO$_2$bz moieties are decomposed with an observed weight loss of 42.83% (calcd. = 44.22%) [59]. Two uncoordinated 2-Mepy moieties undergo decomposition in the temperature range 466–800 °C (obs. = 24.67%, calcd. = 26.12%) [60]. For compound **2**, six coordinated water molecules undergo thermal decomposition in the temperature range 51–115 °C with the observed weight loss of 7.80% (calcd. = 8.50%) [58]. One 2,3-PDCH moiety is decomposed in the temperature range 116–220 °C (obs. = 12.32%, calcd. = 13.14%) [1]. In the third step, three 2,3-PDCH moieties undergo decomposition in 221–455 °C with an observed weight loss of 40.76% (calcd. = 39.42%) [1]. In the temperature range 456–800 °C, the observed weight loss of 24.97% corresponds to the loss of two coordinated 2,3-PDCH moieties (calcd. = 26.28%) [1].

### 3.5. Theoretical Studies

The study of compound **1** was focused on the study of the interesting 2D assembly where the 2-Mepy and 4-NO$_2$bz moieties are held together by C–H···O, $\pi$-$\pi$ and l.p.-$\pi$ interactions. A tetrameric assembly (Figure 7) was used where the carboxylate group of 4-NO$_2$bz has been protonated in order to use a neutral model and estimate the non-covalent contacts free from the influence of strong electrostatic forces. QTAIM and NCI plots, by means of the reduced density gradient (RDG) isosurfaces, were used to characterize the non-covalent contacts in the tetrameric assembly of compound **1**, revealing an intricate combination of interactions; that is, four C–H···O(nitro) interactions that are characterized by bond critical points (BCPs, represented as red spheres) and bond paths (BPs, represented as orange lines) connecting the O-atoms of the nitro groups to two adjacent aromatic H-atoms. The $\pi$-stacking interactions are characterized by two BCPs and BPs interconnecting two C-atoms of the 2-Mepy rings to two C-atoms of the 4-NO$_2$bz moieties.

Interestingly, the QTAIM analysis ratifies the presence of the lp–$\pi$ interaction involving one O-atom of the nitro group. It is characterized by a BCP and BP connecting the O-atom to one C-atom of the 2-Mepy ring. Finally, the QTAIM also confirms the importance of the

C–H⋯π interactions, characterized by a BCP and BP connecting the H and C-atoms. All these interactions are also revealed by green-colored (attractive) RDG isosurfaces. Those characterizing the C–H⋯π and π–π interactions are quite extended, embracing most of the aromatic surfaces. We have evaluated the formation energy of the tetramer, starting from two different (2-Mepy)⋯(4-NO$_2$bz) dimeric assemblies, in order to investigate the relative importance of the H-bonds with respect to the π-based interactions. The dimerization energy starting from the pre-formed π–π dimer (represented in Figure 7a) is –6.2 kcal/mol, which corresponds to the contribution of the four H-bonds (1.55 kcal/mol each H-bond). If the formation of the tetramer is computed starting from the H-bonded dimer (Figure 7b), the binding energy is significantly larger (–14.4 kcal/mol), thus suggesting that the combination of π–interactions (l.p-π, π-π and C–H⋯π) are energetically more relevant than the H-bonds.

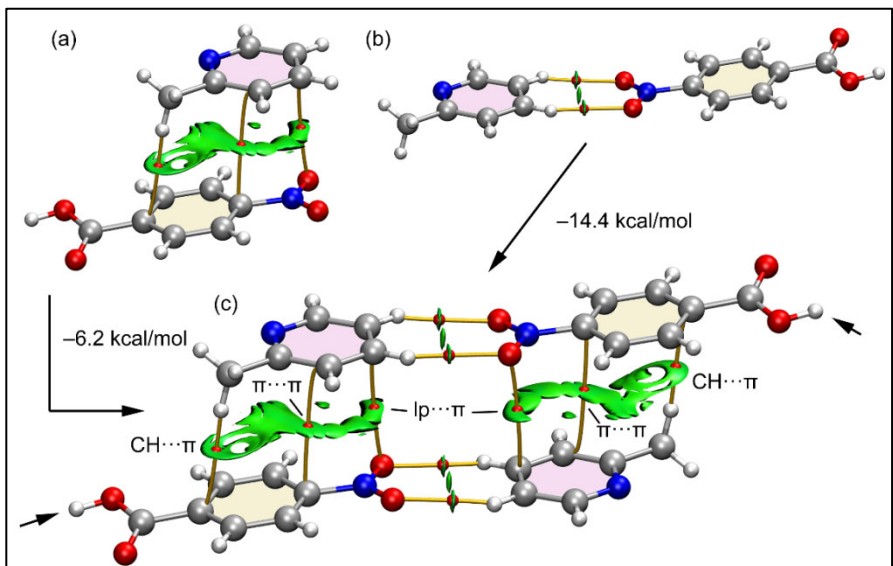

**Figure 7.** Combined QTAIM analysis (BCPs in red, BPs in orange) and NCI surfaces of the (**a**) π-stacked, (**b**) H-bonded dimers of compound **1** and (**c**) the tetramer. The gradient cut-off is ρ = 0.05 a.u., isosurface = 0.5, and the color scale is −0.04 a.u. < (signλ$_2$)ρ < 0.04 a.u. Only intermolecular interactions are shown. The formation energies of the tetramer using the two dimers are also indicated.

For compound **2**, the theoretical study is focused on the analysis of the O–H⋯O H-bonds and π-stacking interactions, which have a significant structural directing role. The joint QTAIM and NCI plot analyses of two H-bonded dimers extracted from the solid state are provided in Figure 8. Each H-bond is characterized by a BCP, BP and dark-blue RDG isosurface connecting the H and O-atoms. The blue NCI Plot color discloses the strong nature of the H-bonds. However, the dimerization energy for the homodimer is positive (+11.3 kcal/mol) due to the electrostatic repulsion (anion⋯anion interaction) and absence of counterions in the model. Consequently, the contribution of the H-bonds was also estimated free from the pure Coulombic repulsion by using the value of the Lagrangian Kinetic energy density (Gr) measured at the bond CP and the equation proposed by Vener et al. (E$_{dis}$ = 0.429 × G$_r$) [61] which was specifically developed for H-bonds in X-ray structures. The values are given in Figure 8 using a red font. The H-bonds are strong, with a dissociation energy of 11.5 kcal/mol each HB, in line with the dark blue color of the RDG isosurfaces. The total dissociation energy of the dimer is 22.6 kcal/mol. Figure 8b shows the QTAIM/NCI Plot analysis of the heterodimer, where the binding energy is very large (–131.6 kcal/mol) due to the strong Coulombic attraction between the counterions. The strength of the H-bonds has been also assessed using the G$_r$ energy predictor to avoid the strong influence of electrostatic forces. In this case, the H-bonds are weaker than those of the homodimer, i.e., 2.2 and 5.1 kcal/mol, thus revealing that the H-bonds between the anionic units are stronger than those between the counterions.

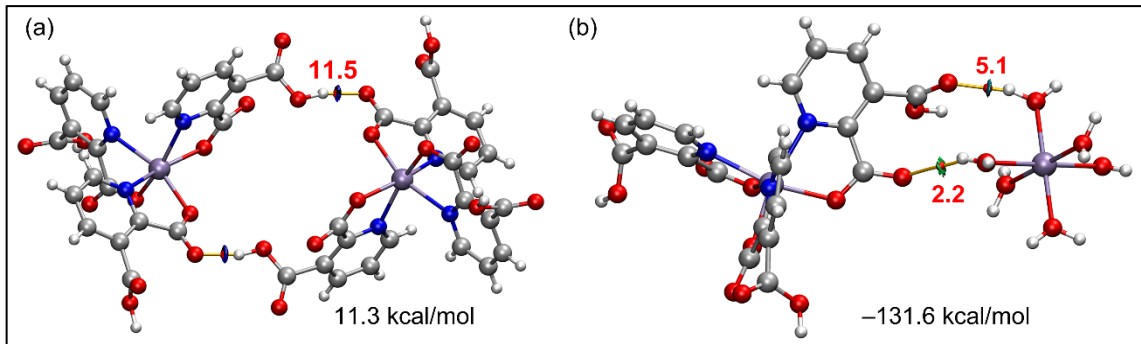

**Figure 8.** Combined QTAIM analysis (bond CPs in red, bond paths in orange) and NCI surfaces of two assemblies of compound **2**, (**a**) one homodimer and (**b**) one heterodimer. The gradient cut-off is $\rho = 0.05$ a.u., isosurfaces = 0.5, and the color scale is $-0.04$ a.u. $< (\text{sign}\lambda_2)\rho < 0.04$ a.u. The dissociation energies of the H-bonds are given in red adjacent to the bond CPs in kcal/mol. The dimerization energies are also indicated.

Finally, we have also analyzed the $\pi$-stacking interaction between the anionic moieties. In this case, to avoid the electrostatic repulsion, two carboxylate groups have been protonated (see small arrows in Figure 9). The $\pi$-stacking interaction is characterized by a large and green RDG isosurface and two BCPs and BPs interconnecting two carbon atoms of the rings. Such a large isosurface reveals a strong complementarity between the aromatic surfaces. The assembly is further stabilized by two symmetric C–H$\cdots$O contacts established between the aromatic –CH groups and the O-atoms of the coordinated carboxylate groups. The strength of these H-bonds is modest ($E_{\text{dis}} = 1.2$ kcal/mol each), thus revealing that the assembly is completely dominated by the $\pi$-stacking interactions, since the total binding energy is $-18.0$ kcal/mol). Such a large binding energy is related to the antiparallel orientation of the aromatic rings and their coordination of the metal centers, which increase the contribution of the dipole$\cdots$dipole interactions.

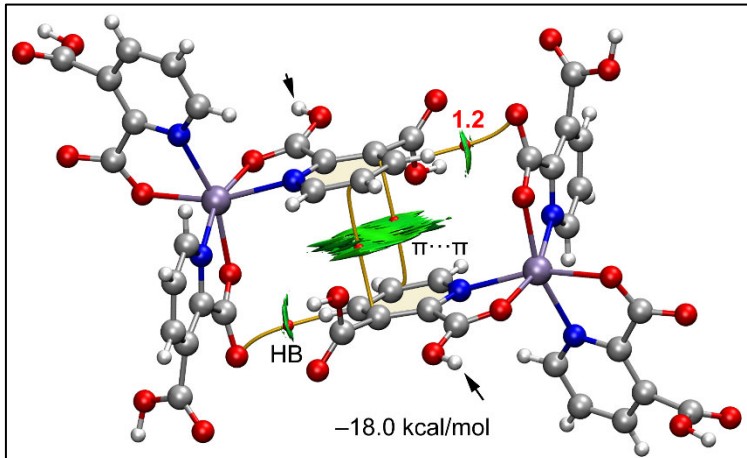

**Figure 9.** Combined QTAIM analysis (BCPs in red, BPs in orange) and NCI surfaces of the $\pi$-stacking assembly of compound **2**. The gradient cut-off is $\rho = 0.05$ a.u., isosurfaces = 0.5, and the color scale is $-0.04$ a.u. $< (\text{sign}\lambda_2)\rho < 0.04$ a.u. The dissociation energies of the H-bonds are given in red adjacent to the bond CPs in kcal/mol. The dimerization energies are also indicated.

## 4. Conclusions

Two Mn(II) multi-component metal–organic compounds, viz. [Mn(H$_2$O)$_6$](2-Mepy)$_2$(4-NO$_2$bz)$_2\cdot$2H$_2$O (**1**) and [Mn(H$_2$O)$_6$][Mn(2,3-PDCH)$_3$]$_2$ (**2**), have been synthesized at room temperature in water medium and characterized using elemental analysis, spectroscopic (FT-IR and electronic), TGA and single-crystal X-ray diffraction techniques. Compound **1**

crystallizes as a multi-component Mn(II) co-crystal hydrate with uncoordinated 2-Mepy, 4-NO$_2$bz and water molecules, while compound **2** is a multi-component metal–organic molecular salt of Mn(II) consisting of cationic [Mn(H$_2$O)$_6$]$^{2+}$ and anionic [Mn(2,3-PDCH)$_3$]$^-$ moieties. Lone-pair (l.p)-$\pi$ and C–H$\cdots\pi$ interactions involving the uncoordinated 2-Mepy and 4-NO$_2$bz moieties of **1** stabilize the layered assembly of the compound. The unconventional inclusion of the guest complex [Mn(H$_2$O)$_6$]$^{2+}$ moiety within the hexameric supramolecular host cavities in compound **2** provides rigidity to the crystal structure. We have carried out DFT calculations, NCI plot analysis and QTAIM computational studies to confirm the energetic features and characteristics of several non-covalent interactions involving the $\pi$-systems (C-H$\cdots\pi$, l.p$\cdots\pi$ and $\pi$-$\pi$) and also the H-bonding interactions (C-H$\cdots$O in **1** and O-H$\cdots$O in **2**). The combination of $\pi$-staking interactions (l.p-$\pi$, $\pi$-$\pi$ and C–H$\cdots\pi$) have higher binding energies, thereby revealing their prominent structure-guiding roles compared to the H-bonds. The large binding energy of $\pi$-stacking interactions in **2** is due to the antiparallel orientation of aromatic rings and their coordination to the metal centers, thereby increasing the contribution of the dipole$\cdots$dipole interactions.

**Supplementary Materials:** The following supporting information can be downloaded at: https://www.mdpi.com/article/10.3390/cryst13050837/s1, Figure S1: Supramolecular 1D Chain of compound **1** assisted by the lattice water molecule; Figure S2: FT-IR spectra of the compounds **1** and **2**; Figure S3: (a) UV-Vis-NIR spectrum of **1**, (b) UV-Vis spectrum of **1**; Figure S4: (a) UV-Vis-NIR spectrum of **2**, (b) UV-Vis spectrum of **2**; Figure S5: Thermogravimetric curves of the compounds **1** and **2**; Table S1: Comparison of crystal parameters of compound **2** with the already reported compound.

**Author Contributions:** Conceptualization, A.F. and M.K.B.; methodology, A.F. and M.K.B.; software, A.F. and R.M.G.; formal analysis, A.F.; investigation, K.K.D.; P.S. and R.M.G.; data curation, M.B.-O.; writing—original draft preparation, K.K.D. and M.K.B.; writing—review and editing, M.K.B.; visualization, A.F.; supervision, M.K.B.; project administration, A.F. and M.K.B.; funding acquisition, A.F. and M.K.B. All authors have read and agreed to the published version of the manuscript.

**Funding:** Financial support was provided by ASTEC, DST, Govt. of Assam (grant number ASTEC/S&T/ 192(177)/2020-2021/43) and the Gobierno de Espana, MICIU/AEI (project number PID2020-115637GB-I00), all of whom are gratefully acknowledged. The authors thank IIT-Guwahati for the TG data.

**Data Availability Statement:** Not applicable.

**Conflicts of Interest:** The authors declare no conflict of interest. The funders had no role in the design of the study; in the collection, analyses, or interpretation of data; in the writing of the manuscript; or in the decision to publish the results.

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
