# Peer review of "Structure Guiding Supramolecular Assemblies in Metal-Organic Multi-Component Compounds of Mn(II): Experimental and Theoretical Studies"

_crystals, doi:10.3390/cryst13050837_

Round 1
Reviewer 1 Report
The authors describe here two crystalline compounds based on the [Mn(H2O)6]2+ dication and containing either nitrobenzoate (compound 1) or Mn(2,3-PDCH)3 (compound 2) as counterions. I am really embarrassed with this manuscript. While the authors made a lengthy introduction based on the supramolecular chemistry, crystal engineering and non-covalent interactions principles, it is clear though that the crystallization of compound 1 was unpredictable, as none of the two potentially coordinating units acted as ligand for Mn(II), but, instead, they co-crystallized. Here the legitimate question which could arise is: did the authors investigate other pyridine derivatives and/or benzoate salts in the same conditions as those for the formation of 1? Then the study would be more complete than a simple description of a serendipitous obtained crystal structure. On the other hand, the crystalline compound 2 was already described in ref [56]. The justification of the authors to describe again in such great detail the crystal structure of 2, such as “Crystal structure of compound 2 has been re-determined with different unit cell parameters having improved R and wR values (TableS1) than the previously reported compound [56].” is far from being convincing. The cell parameters are not so different, and in the original report the quality of the structure was already very good. All this lengthy crystallographic discussion for a compound already described should be removed. It appears so that the crystallographic part of the manuscript is either serendipitous, either reproduction of previously reported results. Then a very interesting theoretical analysis of the two structures is presented, focusing on the non-covalent interactions.
In view of the above arguments, my belief is that the results reported here do not justify in the present form their publication in Crystals.
I have noticed as well several inaccuracies:
1) Page 1 in the Introduction, the sentence “Using organic moieties as building blocks; various research groups…” should be “Using organic moieties as building blocks, various research groups…”. As such, many inappropriate uses of “;” instead of “,” in the middle of a sentence have been noticed. This should be corrected.
2) The formulation of compound 1 as [Mn(H2O)6] 2(2-Mepy) 2(4-NO2bz)·2H2O is totally misleading. It should be [Mn(H2O)6] (2-Mepy)2 (4-NO2bz)2·2H2O. Then it is clear what the composition of the asymmetric unit is.
3) In the structural description of complex 2 it should be also said that Mn2 lies not only on a 3-fold axis but also on an inversion centre.
Author Response
First, we would like to thank this Reviewer for his/her careful reading of the manuscript, important corrections and suggestions. Our point-by-point responses follow:
Reviewer 1
Comments and Suggestions for Authors
Q1. The authors describe here two crystalline compounds based on the [Mn(H2O)6]2+ dication and containing either nitrobenzoate (compound 1) or Mn(2,3-PDCH)3 (compound 2) as counterions. I am really embarrassed with this manuscript. While the authors made a lengthy introduction based on the supramolecular chemistry, crystal engineering and non-covalent interactions principles, it is clear though that the crystallization of compound 1 was unpredictable, as none of the two potentially coordinating units acted as ligand for Mn(II), but, instead, they co-crystallized. Here the legitimate question which could arise is: did the authors investigate other pyridine derivatives and/or benzoate salts in the same conditions as those for the formation of 1? Then the study would be more complete than a simple description of a serendipitous obtained crystal structure.
Reply: We have revised the introduction section in the revised manuscript. Yes, we have investigated other pyridine derivatives and/or benzoate salts in the same conditions as those for the formation of 1; however, we could not crystallize them yet.
Q2. On the other hand, the crystalline compound 2 was already described in ref [56]. The justification of the authors to describe again in such great detail the crystal structure of 2, such as “Crystal structure of compound 2 has been re-determined with different unit cell parameters having improved R and wR values (TableS1) than the previously reported compound [56].” is far from being convincing. The cell parameters are not so different, and in the original report the quality of the structure was already very good. All this lengthy crystallographic discussion for a compound already described should be removed. It appears so that the crystallographic part of the manuscript is either serendipitous, either reproduction of previously reported results. Then a very interesting theoretical analysis of the two structures is presented, focusing on the non-covalent interactions.
In view of the above arguments, my belief is that the results reported here do not justify in the present form their publication in Crystals.
Reply: We have now shortened the description on the crystal structure analysis of compounds 2 and highlighted more on the non covalent interactions representing the unusual crystal packing and the supporting computational studies. We have also removed the misleading sentences of the manuscript as suggested by the esteemed reviewer.
Q3. I have noticed as well several inaccuracies:
1) Page 1 in the Introduction, the sentence “Using organic moieties as building blocks; various research groups…” should be “Using organic moieties as building blocks, various research groups…”. As such, many inappropriate uses of “;” instead of “,” in the middle of a sentence have been noticed. This should be corrected.
Reply: We have done necessary corrections as suggested.
2) The formulation of compound 1 as [Mn(H2O)6] 2(2-Mepy) 2(4-NO2bz)·2H2O is totally misleading. It should be [Mn(H2O)6] (2-Mepy)2 (4-NO2bz)2·2H2O. Then it is clear what the composition of the asymmetric unit is.
Reply: We have modified the formula as suggested
3) In the structural description of complex 2 it should be also said that Mn2 lies not only on a 3-fold axis but also on an inversion centre.
Reply: We have modified the sentence accordingly.
Reviewer 2 Report
The manuscript by Bhattacharyya and co-workers reports the synthesis and detailed characterization of two multi-component coordination compounds of managanese (II) using FTIR and single crystal X-ray analysis. The purity has been confirmed by elemental analysis and the stability has been further evaluated using TGA. The authors have additionally carried out DFT and QTAIM computational analysis to confirm the energetic features and the role of non-covalent interactions. The manuscript has been well written and definitely deserves to be published in crystals. I would appreciate if the authors take into account the following minor points during revision.
1. Since, the structural data of 2 has already been published previously, therefore, it should be mentioned in the abstract and that why the authors have recollected the data and how it differs from the previous report?
2. Authors should add a response to the B-alert in cif file.
3. Please check for little grammatical corrections in the text. For instance, line 57, "which mainly deals" should be "which mainly deal". Line 69, "but also involve" should be instead, "but are also involved" etc.
Author Response
First, we would like to thank this Reviewer for his/her careful reading of the manuscript, corrections and suggestions. Our point-by-point responses follow:
Reviewer 2
Comments and Suggestions for Authors
The manuscript by Bhattacharyya and co-workers reports the synthesis and detailed characterization of two multi-component coordination compounds of managanese (II) using FTIR and single crystal X-ray analysis. The purity has been confirmed by elemental analysis and the stability has been further evaluated using TGA. The authors have additionally carried out DFT and QTAIM computational analysis to confirm the energetic features and the role of non-covalent interactions. The manuscript has been well written and definitely deserves to be published in crystals. I would appreciate if the authors take into account the following minor points during revision.
Q1. Since, the structural data of 2 has already been published previously; therefore, it should be mentioned in the abstract and that why the authors have recollected the data and how it differs from the previous report?
Reply: We have mentioned about compound 2 in the abstract
Q2. Authors should add a response to the B-alert in cif file.
Reply: We have incorporated the response in cif file as suggested.
Q3. Please check for little grammatical corrections in the text. For instance, line 57, "which mainly deals" should be "which mainly deal". Line 69, "but also involve" should be instead, "but are also involved" etc.
Reply: We have done the necessary modifications in the revised manuscript.
Reviewer 3 Report
In the reviewed manuscript M. K. Bhattacharyya et al. reports the synthesis and crystal structures of two Mn(II) multi-component coordination compounds. These compounds have been further characterized using various spectroscopic and analytical techniques. Additionally, the energetical features of the supramolecular assemblies have been explored using DFT calculations, NCI plots and QTAIM computational tools.
In my opinion, the studies done by the Authors are quite well performed and described. Therefore, this contribution could be published in the Crystals Journal after minor revision. Some typos should be corrected, for example:
1. line 198 (TableS1) - Table S1,
2. line 298 (Figure3) - Figure 3,
3. line 413 (Figure7) - Figure 7, and others.
Author Response
First, we would like to thank this Reviewer for his/her careful reading of the manuscript, corrections and suggestions. Our point-by-point responses follow:
Reviewer 3
Comments and Suggestions for Authors
In the reviewed manuscript M. K. Bhattacharyya et al. reports the synthesis and crystal structures of two Mn(II) multi-component coordination compounds. These compounds have been further characterized using various spectroscopic and analytical techniques. Additionally, the energetical features of the supramolecular assemblies have been explored using DFT calculations, NCI plots and QTAIM computational tools.
In my opinion, the studies done by the Authors are quite well performed and described. Therefore, this contribution could be published in the Crystals Journal after minor revision. Some typos should be corrected, for example:
- line 198 (TableS1) - Table S1,
- line 298 (Figure3) - Figure 3,
- line 413 (Figure7) - Figure 7, and others.
Reply: We have done the necessary modifications.
Round 2
Reviewer 1 Report
The authors did appreciable efforts to improve the quality of the manuscript and to present the results, especially with respect to the already published compound 2, in a fairer way. A lot of English polishing is still necessary. Punctuation marks such as “;” are still wrongly used in the middle of sentences instead of commas. The Conclusion paragraph has not been corrected (for example formulae of the compounds). With these additional corrections I think the manuscript can be considered for publication in Crystals especially thanks to the extensive theoretical analysis of the two crystal structures.
Author Response
We thank this referee for his/her second reading of the manuscript, corrections and suggestions. Our "point-by-point" responses are detailed below:
The authors did appreciable efforts to improve the quality of the manuscript and to present the results, especially with respect to the already published compound 2, in a fairer way. A lot of English polishing is still necessary. Punctuation marks such as “;” are still wrongly used in the middle of sentences instead of commas. The Conclusion paragraph has not been corrected (for example formulae of the compounds). With these additional corrections I think the manuscript can be considered for publication in Crystals especially thanks to the extensive theoretical analysis of the two crystal structures.
Reply: Thank you for the comments. We have corrected the formulas in the conclusion section and also changed the wrong punctuation marks ";" by ",".